# Tunable and scalable fabrication of block copolymer-based 3D polymorphic artificial cell membrane array

Dong-Hyun Kang [1,2,5], Won Bae Han [1,3,5], Hyun Il Ryu[1], Nam Hyuk Kim[4], Tae Young Kim[1], Nakwon Choi [1,3], Ji Yoon Kang[1], Yeon Gyu Yu[4] & Tae Song Kim [1✉]

Owing to their excellent durability, tunable physical properties, and biofunctionality, block copolymer-based membranes provide a platform for various biotechnological applications. However, conventional approaches for fabricating block copolymer membranes produce only planar or suspended polymersome structures, which limits their utilization. This study is the first to demonstrate that an electric-field-assisted self-assembly technique can allow controllable and scalable fabrication of 3-dimensional block copolymer artificial cell membranes (3DBCPMs) immobilized on predefined locations. Topographically and chemically structured microwell array templates facilitate uniform patterning of block copolymers and serve as reactors for the effective growth of 3DBCPMs. Modulating the concentration of the block copolymer and the amplitude/frequency of the electric field generates 3DBCPMs with diverse shapes, controlled sizes, and high stability (100% survival over 50 days). In vitro protein–membrane assays and mimicking of human intestinal organs highlight the potential of 3DBCPMs for a variety of biological applications such as artificial cells, cell-mimetic biosensors, and bioreactors.

[1] Creative Research Center for Brain Science, Korea Institute of Science and Technology, 5, Hwarang-ro 14-gil, Seongbuk-gu, Seoul 02792, Republic of Korea. [2] Micro Nano Fab Center, Korea Institute of Science and Technology, 5, Hwarang-ro 14-gil, Seongbuk-gu, Seoul 02792, Republic of Korea. [3] KU-KIST Graduate School of Converging Science and Technology, Korea University, 145 Anam-ro, Seongbuk-gu, Seoul 02841, Republic of Korea. [4] Department of Chemistry, Kookmin University, 77 Jeongneung-ro, Seongbuk-gu, Seoul 02707, Republic of Korea. [5]These authors contributed equally: Dong-Hyun Kang, Won Bae Han. ✉email: tskim@kist.re.kr

Self-assembly of block copolymers has attracted considerable interest in fundamental research and technical applications because of the exceptional morphological and structural versatility. In the bulk state, block copolymers undergo micro-phase separation by a driving force that minimizes the free energy of the system to form various structures, such as spheres, cylinders, and lamellae[1], providing the capability to generate architectural complexity and periodicity that are difficult to achieve with conventional synthetic or lithographic technologies[2,3]. This behavior has been widely exploited in 2-dimensional (2D)/3-dimensional (3D) nanoscale patterning for functional devices with high throughput and fine feature sizes[4–6], porous membranes for molecular separation and water purification[7,8], and other advanced material designs[9–11]. The spin-coating method is widely used to form block copolymer thin films on templates; however, it is restricted in terms of producing patterned areas, which limits its utility in complicated and sophisticated device fabrication. Various methods, such as chemical[12,13] or topographical patterning[14], solvent annealing[15], and techniques based on electric fields[16], thermal gradients[17], and shear forces[18] have been developed to address this issue and provide ordered orientation. However, among these methods, only chemical or topographical patterning is effective in achieving patterned areas with predefined sizes and geometries at the desired locations.

Owing to their superior mechanical/chemical stability, adjustable membrane permeability, and various surface functionalities realized through polymer engineering, block copolymer membranes have considerable potential for biological applications. In particular, they are promising alternatives to phospholipid-based membranes as artificial cell membranes for elucidating the underlying mechanical/biological mechanisms of cells and providing a platform for bioreactors and biosensors. However, in aqueous media, block copolymers spontaneously self-assemble into only one type of energetically favorable structure depending on the packing parameter ($p = v/a_0 \, l_c$, where $v$ and $l_c$ are the volume and length of the hydrophobic domain, respectively, and $a_0$ is the optimal area of the hydrophilic domain)[19]. Moreover, their polymer chains are difficult to rearrange because their molecular weight is greater than that of phospholipids, limiting their applications. Although osmolarity control[20], chemical addition[21], liquid-crystalline component incorporation[22], and deoxyribonucleic acid (DNA)-directed self-assembly[23] have been reported to induce morphological transitions, most approaches produce irreversible shapes under specific conditions or are time-consuming. Therefore, there is a great need to overcome the aforementioned energetic penalties to accomplish a facile, systematic shape transformation.

Herein, we demonstrate a method for fabricating 3D block copolymer artificial cell membranes (3DBCPMs) by combining the top-down control of the feature sizes and locations of 3DBCPMs via micro-contact printing and lithographic techniques with the bottom-up control of the diameter/length and shapes of 3DBCPMs via electric field-assisted molecular self-assembly (Fig. 1a). Topographically and chemically structured microwell array templates not only allow micropatterning of block copolymers but also serve as confined reactors where block copolymers self-assemble and fuse to produce 3DBCPMs. An electric field with controlled amplitude and frequency regulates the morphology of 3DBCPMs over a packing parameter-dependent structure. To date, no method has been reported for the scalable fabrication ($\sim$300,000/cm$^2$) of microscale 3D block copolymer structures immobilized on patterned templates with morphological versatility, controlled size, and extremely long lifetime (100% survival over 50 days). Although simple rehydration[24], sonication[25], solvent annealing[26], and electroformation[27,28] methods have been adopted to produce 3D block copolymer structures, most of these methods produce only suspended vesicles with large size polydispersity, which are difficult to immobilize onto substrates for further experimentation and observation.

## Results

**3D block copolymer structures on Si microwell arrays.** Figure 1b shows a 3D fluorescence image reconstructed from 2D slices of a representative 3DBCPM comprising an amphiphilic diblock copolymer, polybutadiene-*b*-polyethyleneoxide (PBd-PEO). It clearly shows extremely long structures with similar shapes, standing along the z-axis of the template. Cross-sectional transmission electron microscopy (TEM) explicitly determined the unilamellar nature of such structures, which was confirmed by the thickness of PBd-PEO bilayer ($\sim$10 nm)[29] (Fig. 1c). This implies that 3DBCPMs exhibit biofunctionality and thus have the potential for various biological applications ranging from drug screening and biological assays to biosensing.

The principal mechanism for the fabrication of 3DBCPMs relies on a combination of patterning of block copolymers into topographically and chemically defined microwell arrays using the dewetting process and self-assembly of block copolymers in aqueous solution with the aid of an electric field (Fig. 2a). An array of microwells fabricated on a silicon (Si) template provides confined spaces for both the isolation of block copolymers and the fusion of swollen block copolymer layers (Supplementary Fig. S1), which has been proven in our previous reports[30]. The micro-contact printing technique deploys a self-assembled monolayer (SAM) of 1H,1H,2H,2H-perfluorododecyltrichlorosilane (PFDDTS) on the template surface treated with UV/ozone to modulate the surface property to hydrophobic/oleophobic, where the block copolymer solution completely dewets. After printing for 6 min, SAM covers the entire surface, as confirmed by atomic force microscopy (AFM) and contact angle measurements (Supplementary Fig. S2). The spin-casting technique facilitates perfect patterning of block copolymers into microwells with negligible residues on the surface (Supplementary Fig. S3), allowing us to determine the surface area of block copolymer lamellae in individual microwells and hence the final size of 3DBCPM (Supplementary Fig. S4). Other silanes that have shorter fluorocarbon chains than PFDDTS are inappropriate for the dewetting process because of their relatively higher surface energies (Supplementary Fig. S5), leading to poor patterning. Upon hydration with an aqueous solution, microphase separation between the hydrophilic and hydrophobic domains in PBd-PEO forms hexagonal hydrophobic rods surrounded by a hydrophilic layer (<10% hydration)[31] (Supplementary Fig. S6). Further hydration results in the fusion of the rods into lamellae, followed by detachment of the lamellae owing to steric repulsion between the facing hydrophilic layers. These processes are continuously repeated layer by layer to form a 3DBCPM. Here, an electric field with different amplitudes and frequencies facilitates the fusion, growth, and transformation of 3DBCPM in a controlled manner. One of the main shapes of 3DBCPM is a unilamellar spherical structure (Fig. 2b). This structure is rooted in the microwell and retains structural integrity with slight undulations in a gentle flow. Block copolymer concentrations of 1.5, 2.0, and 2.5 wt% resulted in different sizes, with average diameters of 12.2, 14.8, and 16.0 μm, respectively, and a coefficient of variation of less than 13.5% (Fig. 2c and Supplementary Fig. S7). It is worth noting that several tens of microscale block copolymer structures with such a narrow size distribution could hardly be realized with the previously reported fabrication methods, such as gentle rehydration, sonication, solution exchange, and electroformation. Furthermore, if we simplify the formation mechanism, where 3DBCPM grows until the block copolymers deposited in a

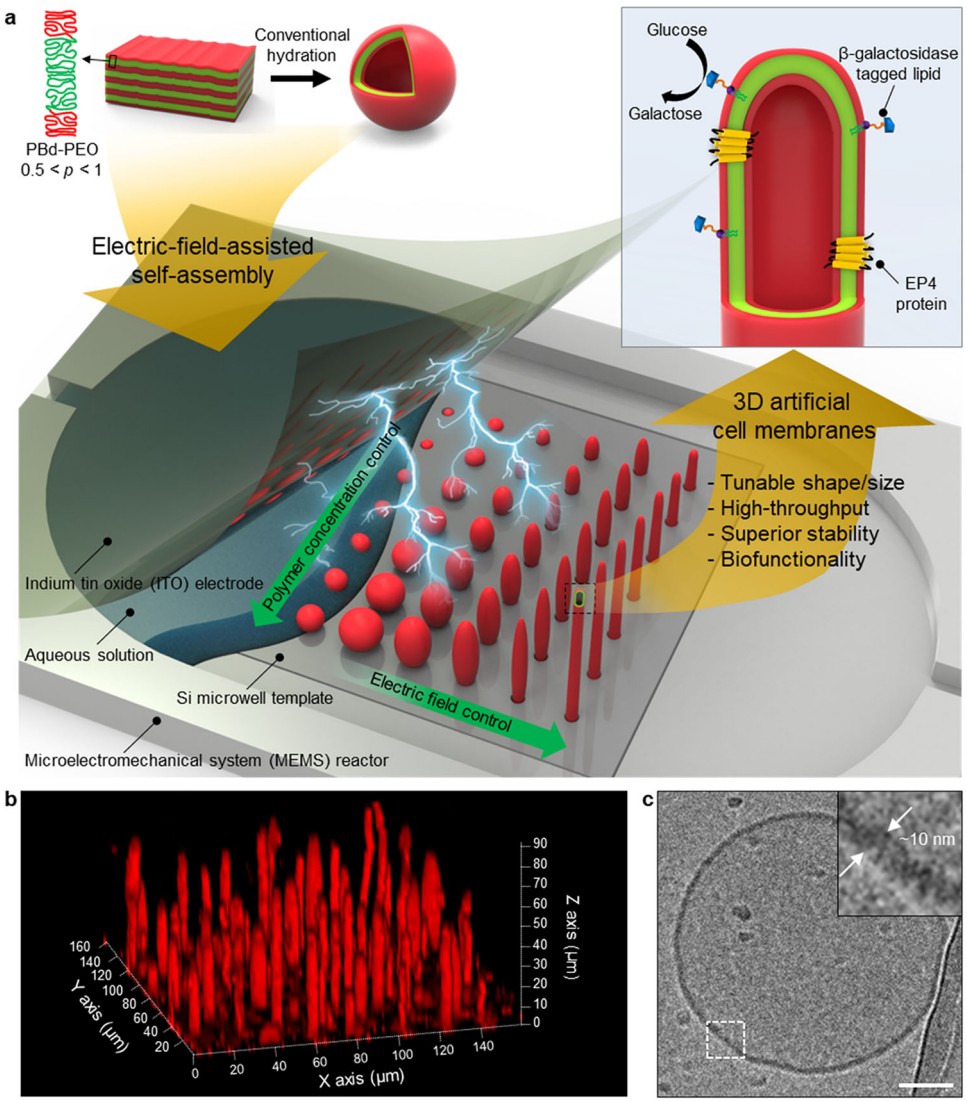

**Fig. 1 3D block copolymer artificial cell membranes (3DBCPMs) for versatile biological applications. a** Electric-field-assisted self-assembly method for the scalable fabrication of 3DBCPMs with diverse shapes/sizes, high stability, and biofunctionality on silicon (Si) microwell template using a diblock copolymer, namely polybutadiene-*b*-polyethyleneoxide (PBd-PEO), as a building block and representative biological applications such as protein–membrane assay using prostaglandin E2 receptor 4 (EP4) and artificial intestinal organ using β-galactosidase tagged lipid. The packing parameter, *p*, is an inherent value of polymers associated with the volume and length of the hydrophobic domain and the optimal area of the hydrophilic domain. **b** A fluorescence microscopy image of a cilia-like long 3DBCPM array produced from a patterned template. **c** A cross-sectional transmission electron microscopy (TEM) image of a unilamellar 3DBCPM composed of ~10-nm-thick bilayer membrane. Scale bar: 100 nm.

microwell are exhausted, we can determine the diameter of 3DBCPM as $d = 1.14 + 8.7x - 1.3x^2 + 0.1x^3$ ($x$ denotes the block copolymer concentration, Supplementary Fig. S4), which highlights the strong size-selection capability of the proposed approach. A cilia-like elongated structure is another main 3DBCPM shape that can be generated by tuning the electric field (Fig. 2d). Such a structure is thinner than the spherical structure and grows upright in a vertical direction from the microwell. The average lengths were 27.4, 30.8, 38.6, and 69.1 μm for block copolymer concentrations of 1.5, 2.0, 2.5, and 4.0 wt%, respectively (Fig. 2e). We suppose that the structure could grow longer than ~100 μm if both the amount of block copolymer and the height of the microfluidic channel in the microelectromechanical system (MEMS) reactor used in this experiment (detailed structure in Supplementary Fig. S8) were sufficiently large. In addition to the aforementioned spherical and cilia structures, reversible, diverse shapes with different curvatures from 1.1 to 9.5 μm can be produced by controlling the electric field (Fig. 2f, and

Supplementary Fig. S9) (the detailed fabrication mechanism is discussed in the next paragraph). Compared with conventional methods that focus on size or shape control, our strategy satisfies both size and shape tunabilities. In addition, a large-scale, uniform array of 3D block copolymer structures that are stably immobilized on predefined locations has not been reported thus far; therefore, this unique system could pave the way for future biological platforms in drug screening, biosensors, bioreactors, and cellular and molecular life sciences.

**Morphology dynamics of 3DBCPM under an electric field.** A combination of three factors, namely the concentration of block copolymer and the frequency and amplitude of electric field, modulates the size and shape of 3DBCPM. To gain deeper insights into how such factors affect the formation of 3DBCPM, we employed the Maxwell–Wagner model (Fig. 3a). The morphology dynamics of membranes in response to an electric field

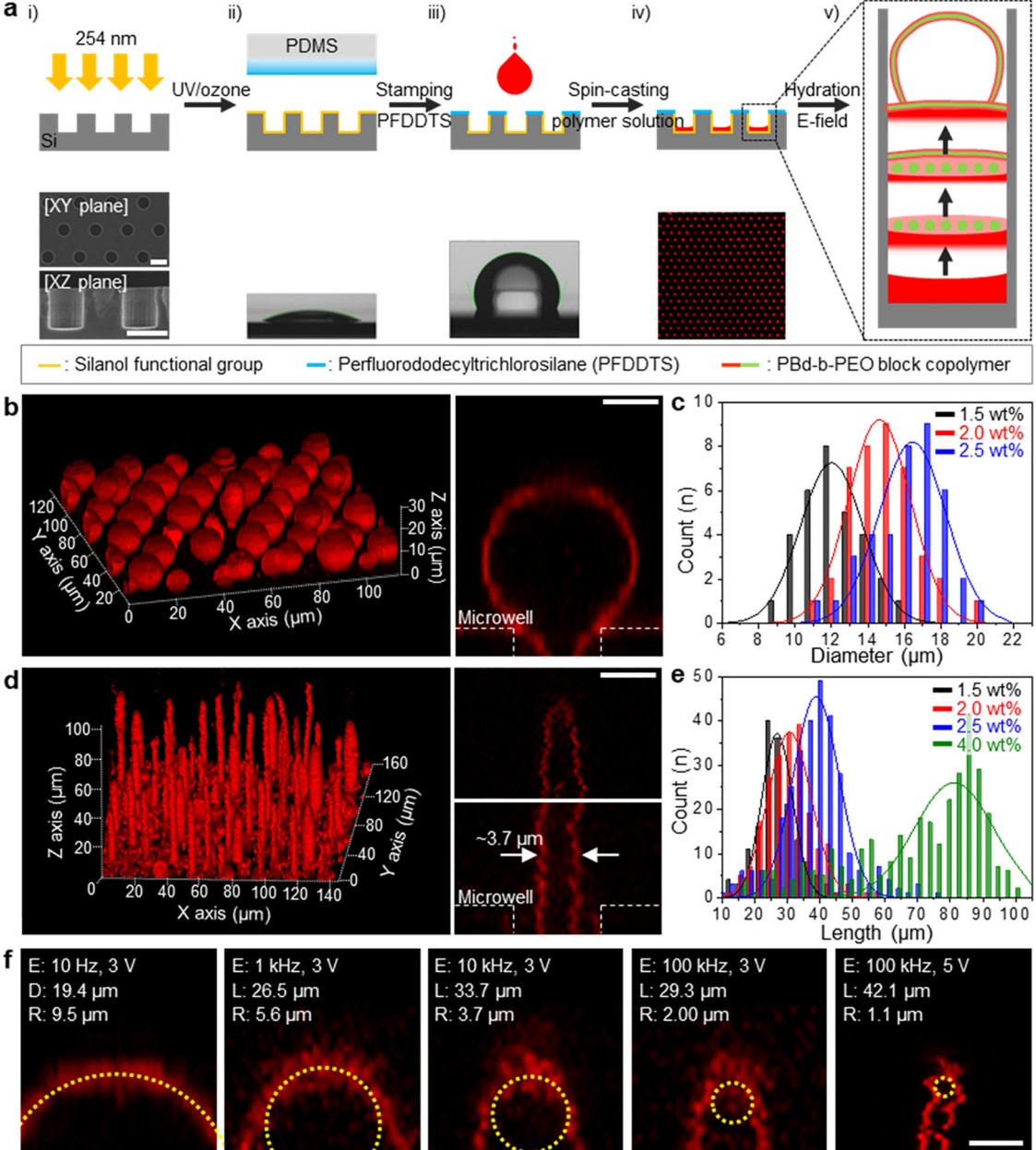

**Fig. 2 Formation of uniform 3DBCPM arrays on a topographically and chemically structured Si microwell template. a** Schematic illustration of fabrication steps for 3DBCPMs and corresponding images: (i) surface termination with hydroxyl groups via ultraviolet (UV)/ozone treatment and scanning electron microscopy (SEM) images of Si microwell template; (ii) formation of fluorocarbon-based self-assembled monolayer (SAM) via micro-contact printing and contact angle measurement of the UV/ozone treated template; (iii) isolation of hydrophilic microwells by the hydrophobic SAM region and CA measurement of the SAM-treated template; (iv) patterning of amphiphilic block copolymer domains via spontaneous dewetting and a fluorescence microscopy image of the block copolymer-coated template; and (v) sequential processes for the formation of 3DBCPM through electric field-assisted self-assembly during hydration. Scale bar: 10 μm. **b** Confocal fluorescence microscopy image of spherical 3DBCPM array (left) and its magnified cross-sectional view (right). Scale bar: 5 μm. **c** Size distributions of spherical 3DBCPMs produced with different block copolymer concentrations. **d** Confocal fluorescence microscopy image of cilia-like 3DBCPM array (left) and its magnified cross-sectional view (right). Scale bar: 5 μm. **e** Length distributions of cilia-like 3DBCPMs produced with different block copolymer concentrations. **f** 3DBCPM with diverse shapes and different curvatures produced from 2.5 wt% of PBd-PEO under different conditions, where E, D, L, and R represent the electric field and the diameter, length, and radius of 3DBCPM, respectively. Scale bar: 5 μm.

applied between indium tin oxide (ITO) and a Si template relate to two different time scales, the Maxwell–Wagner time scale ($t_{MW}$) and charge-on time scale ($t_c$)[32]:

$$t_{MW} = \frac{\varepsilon_{in} + 2\varepsilon_{ex}}{\lambda_{in} + 2\lambda_{ex}}, \quad t_c = RC_m\left(\frac{1}{\lambda_{in}} + \frac{1}{2\lambda_{ex}}\right) \quad (1)$$

where $\varepsilon_{in}$ and $\varepsilon_{ex}$ are the permittivities of the interior and exterior fluids, respectively; $\lambda_{in}$ and $\lambda_{ex}$ are the conductivities of the interior and exterior fluids, respectively; $R$ is the radius of 3DBCPM, and $C_m$ is the dimensionless membrane capacitance per unit area. An electrically neutral membrane becomes polarized on $t_{MW}$ and charged on $t_c$. At low frequencies, a cycle ($t_o$) that is sufficiently longer than $t_c$ allows a membrane to be charged

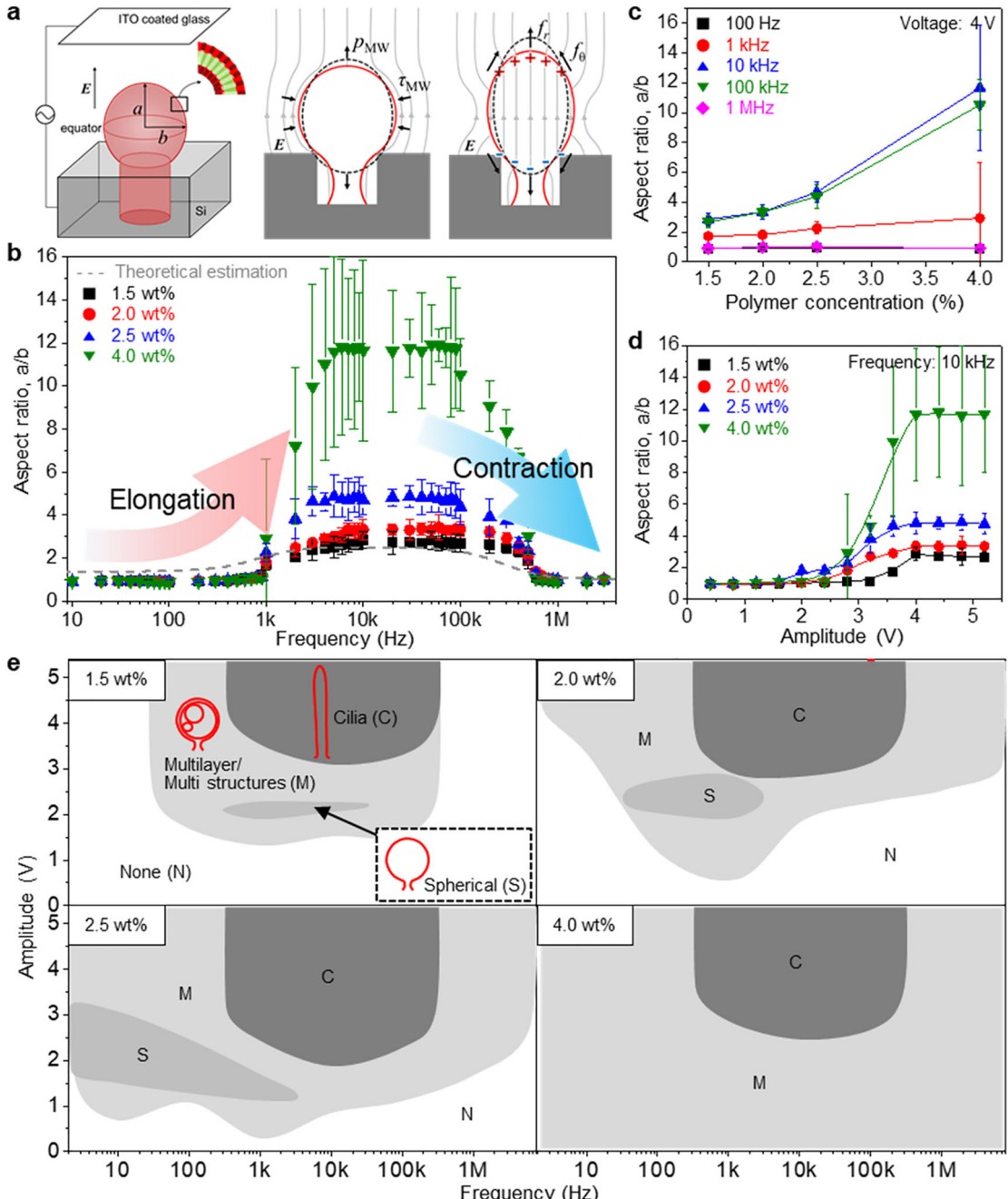

**Fig. 3 Morphology dynamics of a 3DBCPM under an electric field with different amplitudes and frequencies. a** Mechanism for the morphological transformation of a 3DBCPM related to net charge distributions on the membrane of the 3DBCPM in an electric field, where $\tau_{MW}$, $p_{MW}$, $f_\theta$, and $fr$ denote the radial Maxwell stress and pressure and the tangential and normal forces, respectively. **b** Theoretical modeling and experimental observation for the frequency-dependent morphology dynamics of 3DBCPM. All 3DBCPMs were produced for an identical time of an hour (except for the time to increase an amplitude by 100 mV every 5 min until the amplitude reached a desired value). Data are presented as means ± standard deviation ($n = 5$ independent samples). **c**, **d** Changes in the aspect ratio of the 3DBCPM as a function of **c** block copolymer concentrations and **d** electric field amplitude. Data are presented as means ± standard deviation ($n = 5$ independent samples). **e** Morphological phase diagram of the 3DBCPM associated with three factors: block copolymer concentration and frequency and amplitude of the electric field.

and act as a capacitor such that the large impedance of the membrane obstructs the penetration of the electric field through the 3DBCPM. Consequently, the 3DBCPM is squeezed at the equator and slightly pulled along the poles by the radial Maxwell stress ($\tau_{MW}$) or pressure ($p_{MW}$) that arises from a tangential electric field. At intermediate frequencies, the 3DBCPM becomes polarized but charged owing to $t_o$, which is longer than $t_{MW}$ but shorter than $t_c$, allowing an electric field to penetrate the

3DBCPM during $t_o$ and producing tangential and normal forces ($f_\theta$ and $f_r$). These forces lead to gradual elongation of the 3DBCPM along the poles. By contrast, the influence of the electric field is negligible at high frequencies because $t_o$ is too short for the 3DBCPM to be polarized and charged[33,34]. According to this theory, the 3DBCPM should have a spherical shape at a frequency of <1 kHz and >1 MHz and an elongated shape at a frequency between 1 kHz and 1 MHz (Fig. 3b). Here, we defined 3DBCPMs

with aspect ratios below and above 1.5 as 'spherical shape' and 'cilia', respectively, for structural distinction. The experimental results (aspect ratio, <1.1) are nearly consistent with such a theoretical estimation (aspect ratio, <1.3). Note that the theoretical estimation was based on the electro-deformation model of the closed vesicle; thus, there was a difference in the degree of elongation, i.e., aspect ratio, in the range of 1 kHz to 1 MHz between the estimation and our system (detailed description in Supplementary information). The aspect ratio sharply increased in the frequency range of 1 kHz to 1 MHz as the block copolymer concentration increased from 1.5 wt% to 4.0 wt% (Fig. 3c), implying that the block copolymer concentration plays an important role in determining the aspect ratio of 3DBCPM. In addition, the amplitude of electric field strongly affects the aspect ratio (Fig. 3d). A low amplitude (<~2.5 V) could not induce elongation of 3DBCPM, whereas at a sufficiently high amplitude (>~2.5 V), the aspect ratio started to increase and became saturated at different levels as a function of the block copolymer concentration. We suppose that the saturated aspect ratio is associated with the total amount of block copolymers deposited in the microwell. Figure 3e shows a comprehensive relationship between the shape of 3DBCPM and the three above-mentioned factors (detailed data in Supplementary Fig. S10). At 1.5 wt%, the total area of 3DBCPM, including the cilia, multilayer/multiple, and spherical structures, is the smallest among all the concentrations owing to limited block copolymer sources. As the concentration increases, all the areas become wider, while a high concentration (4.0 wt%) results in an area that is too wide with unwanted/uncontrollable multilayer/multiple structures and also eliminates the area for spherical structures. We suppose that a microwell could not accommodate and confine block copolymer structures rapidly and continuously produced from highly concentrated block copolymers; hence, the structures protruded out of the microwell without having sufficient time to merge into a larger unilamellar structure. Indeed, fusion of membranes occurs when two facing membranes become sufficiently close to deplete water molecules between them and thus make their structures energetically unstable/unfavorable[35–37]. In other words, the proposed template is a unique system in which a microwell geometrically plays a vital role as a fusion reactor for effectively producing uniform, giant, unilamellar 3DBCPMs. Although a spontaneous rehydration method to fabricate block copolymer spherical structures from chemically patterned templates has been reported[13], the fabrication of giant (>10 μm in diameter), unilamellar structures with narrow size distributions and diverse shapes has not been considered. The area for the cilia is in a narrow frequency range, i.e., from 1 kHz to 100 kHz, and the minimum amplitude required for the cilia decreases as the block copolymer concentration increases, whereas it increases again at 4.0 wt%. In addition to elucidating the mechanism of 3DBCPM formation, the proposed morphological phase diagram allows us to systematically produce 3DBCPMs with any electrically neutral block copolymer of interest.

**Mechanical stability of hydrogel-supported 3DBCPM.** In addition to size and shape tunability, the structural and mechanical robustness of 3DBCPMs is a prerequisite for their practical use as biological platforms. However, the generated 3DBCPMs slowly transform their original morphology into a different morphology in response to different amplitudes and frequencies. Furthermore, unlike spherical structures, cilia structures grow along the direction of the electric field and maintain their morphology under the electric field; hence, removal of the electric field causes morphological changes and makes the structures vulnerable to external environments. To circumvent

these issues and ensure not only structural integrity but also water permeability for solution exchange in further bioassays, we used a hydrogel as a support (Fig. 4a). Interestingly, the injection of the hydrogel solution decreased the aspect ratio but increased the diameter (Fig. 4b). We suppose that this phenomenon originates from the reduced amplitude of electric field, possibly owing to impurities or ions present in the hydrogel solution. Despite the slight changes in morphology, the final structure still had a high aspect ratio and excellent mechanical stability regardless of the electric field conditions. When the electric field disappeared, the hydrogel-supported 3DBCPM showed negligible undulations under a mild flow in contrast to the 3DBCPM without the hydrogel, as confirmed by time-sequential images (Fig. 4c). To quantitatively evaluate the mechanical robustness of the hydrogel-supported 3DBCPM, we applied various pressures to the 3DBCPM by controlling the flow rate of the aqueous solution (Fig. 4d). As the hydrogel concentration increased, the mechanical modulus of the 3DBCPM increased linearly, indicating that the addition of the hydrogel effectively prevents structural deformations under external mechanical stimuli with a pressure of several kilopascals. Furthermore, the addition of the hydrogel improved the lifetime of the 3DBCPM from ~80% survival to ~100% survival over 50 days, which is approximately one order of magnitude longer than the lifetime of lipid-based structures (up to 5.5 days)[38,39] (Fig. 4e). This suggests that hydrogel-supported 3DBCPMs with exceptional structural and mechanical robustness are potential alternatives to fragile, unstable artificial lipid cell membranes for mimicking cellular structures and implementing diverse biological assays.

**Biological applications of 3DBCPM as artificial cell membrane.** As a proof of concept, we performed two experiments: a protein–membrane interaction assay and mimicking of human intestinal organs, using hydrogel-supported structures without e-field. As the integration of membrane proteins into artificial membranes is an essential step for model cell membrane assays and cell-mimetic biosensors/bioreactors[40], the interaction of 3DBCPM with proteins needs to be investigated. Therefore, we incorporated a representative G protein-coupled receptor, human prostaglandin E2 receptor 4 (EP4), which is involved in pathological and physiological responses, into 3DBCPMs. Here, amphipathic poly-γ-glutamic acid (APG) and green fluorescent protein (GFP) were used for stabilization/reconstitution and visualization of the protein, respectively. APG has been proven to stabilize membrane proteins in their active conformations in aqueous solution and assist them in incorporation into lipid membranes while maintaining the integrity of the membrane[41,42]. APG-stabilized proteins were bound and inserted into a 3DBCPM presumably by hydrophobic interactions between the hydrophobic regions of the 3DBCPM and the nonpolar alkyl chains of APG, and spontaneous dissociation of the APG completed the reconstitution process to leave the protein in the 3DBCPM (Fig. 5a). Upon the introduction of APG-stabilized EP4, green fluorescence appeared in the 3DBCPM (Fig. 5b). The obvious difference in fluorescence intensity before and after the addition of EP4, even after thorough washing and pH shock steps, confirmed the capability of the 3DBCPM to incorporate with membrane proteins (Fig. 5c and Supplementary Fig. S11). Various enzymes, such as maltase-glucoamylase, sucrase-isomaltase, and lactase, present on the surface of human intestinal cilia decompose polysaccharides into monosaccharides to improve nutrient absorption[43]. We mimicked these human intestinal organs by incorporating a commercially available enzyme, i.e., bacterial lactase (beta-galactosidase, β-Ga), into the cilia developed in this work (Fig. 5d). β-Ga-conjugated lipids embedded in

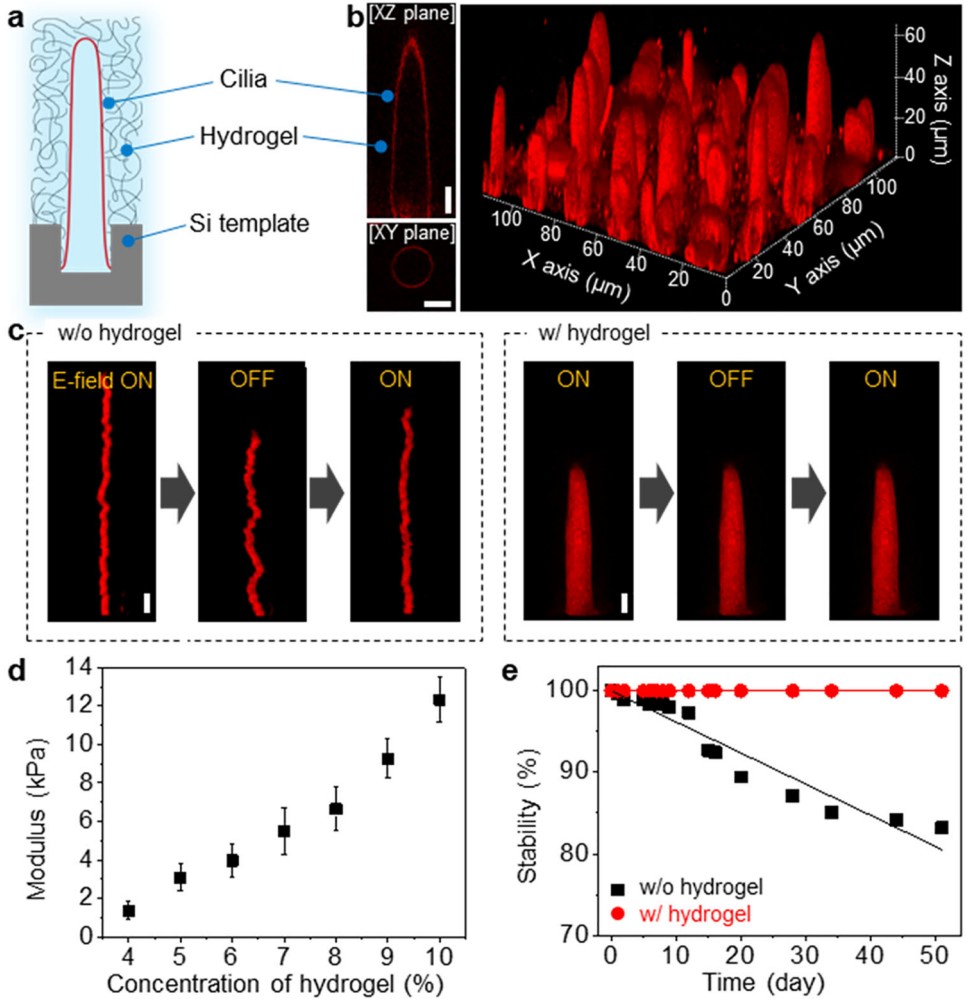

**Fig. 4 Structural and mechanical properties of a cilia-like 3DBCPM supported with a hydrogel. a** Schematic illustration of a cilia-like 3DBCPM with a hydrogel as a support. **b** Confocal fluorescence microscopy images of a hydrogel-supported 3DBCPM array. Scale bar: 5 μm. **c** Time-sequential images of cilia-like 3DBCPMs with and without hydrogel support under a mild flow of aqueous solution. Scale bar: 5 μm. **d** Mechanical modulus of 3DBCPMs with different hydrogel concentrations. Data are presented as means ± standard deviation ($n = 4$ independent samples). **e** Mechanical stability of 3DBCPMs with and without hydrogel support in aqueous solution.

the 3DBCPM at 1 mol% hydrolyzed fluorescein di-β-D-galacto-pyranoside (FDG) into galactose and GFPs, leading to an increase in fluorescence. We compared the fluorescence intensities of cilia, spherical, and planar structures to evaluate the efficacy of the enzymatic reactions (Fig. 5e). Overall, the cilia structures showed the most rapid increase in fluorescence intensity. We suppose that the large surface area of the cilia structures accommodates much more β-Ga compared to the other structures and reacts with FDG frequently. This phenomenon is coincident with the fact that the cilia that exist in visual, olfactory, gustatory, digestive, and nerve systems have distinctive features of protruding, elongated 3D architectures with a large surface area, and coupled with proteins and other biomolecules, they perform essential physiological tasks in the body, such as sensing, communication, and metabolism[44,45]. The reaction rate constant can be estimated according to the first-order reaction, $P = P_0 e^{kt}$, where P is the concentration of the reactant, $P_0$ is the initial concentration of the reactant, k is the rate constant, and t is the time. The cilia had the highest reaction rate constant, even when the rate constant was normalized by the surface area (Fig. 5f). We suppose that this behavior originates from a 3D architecture favorable for an active and effective reaction as well as from the aforementioned large surface area. This conclusion was confirmed by the

Michaelis–Menten equation (Fig. S12), highlighting the potential of 3DBCPMs, especially cilia, as highly reactive platforms for various applications in the fields of artificial organs, biosensors, and drug screening.

## Discussion

In summary, we developed a facile and systematic approach for the formation of diblock copolymer-based 3D structure arrays immobilized on a Si template. Along with template-guided self-assembly, control of the electric field produces structures with various shapes, uniform size, superior stability, and biofunctionality, which can replace polymorphic, but fragile lipid-based structures and simple polymersomes, thereby enhancing the potential of artificial cell membranes for various applications. The proposed strategy may be developed further by integrating electrodes into the template, employing other block copolymers, or incorporating other membrane proteins and biomolecules.

## Methods

**Materials**. Si wafers (p-type, 100) were purchased from Hissan (South Korea), and AZ GXR-601 (46 cps) photoresist (PR) was purchased from Merck (Germany). Poly-butadiene-*b*-polyethyleneoxide (PBd-PEO; Mn: 1200 for PBd and 600 for PEO) and rhodamine-terminated PBd-PEO were purchased from Polymer Source (Canada), and

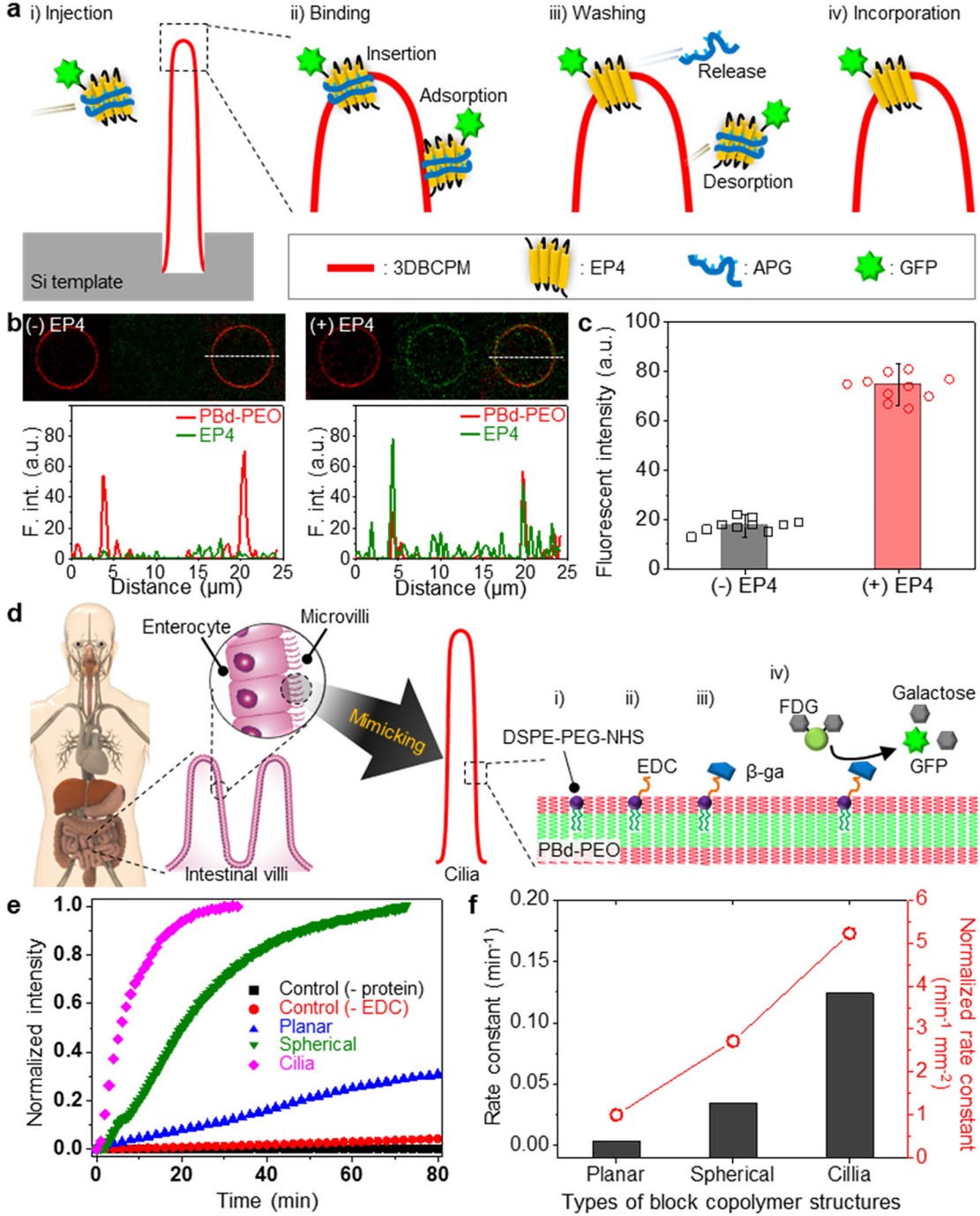

**Fig. 5 Two representative biological applications based on a 3DBCPM as a platform. a** Study of protein–membrane interaction by incorporating EP4 proteins that were conjugated with a green fluorescent protein (GFP) and stabilized with amphipathic poly-γ-glutamic acid (APG): (i) injection of EP4; (ii) Insertion and adsorption of EP4 to 3DBCPM; (iii) Release of APG from 3DBCPM and desorption of adsorbed EP4 by repeated washing; and (iv) Successful incorporation of EP4 to 3DBCPM. **b** Cross-sectional confocal fluorescence microscopy images (top) and fluorescence intensities (bottom) of 3DBCPMs without (left) and with (right) EP4 incorporation. **c** Confirmation of the protein reconstitution on the 3DBCPM by distinguishing groups of fluorescence intensities of the 3DBCPM membranes incorporated without and with EP4. Data are presented as means ± standard deviation ($n = 10$ independent samples). **d** Fabrication and working principles of 3DBCPM-based artificial intestinal organs: (i) introduction of 1 mol% of 1,2-distearoyl-sn-glycero-3-phosphoethanolamine-*N*-[succinimidyl(polyethylene glycol)] (DSPE-PEG-NHS) onto the 3DBCPM; (ii) coupling of NHS with 1-ethyl-3-(3-dimethylaminopropyl)carbodiimide (EDC); (iii) conjugation of beta-galactosidase (β-Ga); and (iv) enzymatic reaction where fluorescein di-β-D-galactopyranoside (FDG) is hydrolyzed into galactose by β-Ga, releasing GFP molecules that cause an increase in fluorescence. **e** Comparison of time-dependent enzymatic kinetics of the proposed spherical and cilia-like 3DBCPMs and a planar block copolymer structure by monitoring the normalized fluorescence intensities after the enzymatic reaction. **f** Estimated enzymatic reaction rate constants and surface-area-normalized rate constants of three different block copolymer structures: planar, spherical, and cilia.

1,2-distearoyl-sn-glycero-3-phosphoethanolamine-N-[carboxy(polyethylene glycol)-2000] (DSPE-PEG2000-COOH) was purchased from Avanti Polar Lipids (USA). Polydimethylsiloxane (PDMS) was purchased from Dow Corning (USA) and used by mixing the base and curing agent at a ratio of 10:1. Hexamethyldisilazane (HMDS), perfluorooctyltrichlorosilane (PFOTS), perfluorodecyltrichlorosilane (PFDTS), perfluorododecyltrichlorosilane (PFDDTS), poly(ethylene glycol) diacrylate (PEGDMA; Mn: 1000), 2-hydroxy-2-methyl-propiophenone, poly(ethylene glycol)-block-poly(-propylene glycol)-block-poly(ethylene glycol) (Poloxamer 188) solution, 4-(2-hydroxyethyl)-1-piperazineethanesulfonic acid (HEPES), 1-ethyl-3-(3-dimethylaminopropyl) carbodiimide (EDC), fluorodeoxyglucose (FDG), sucrose, isopropyl β-D-1-thiogalactopyranoside (IPTG), sarkosyl, β-mercaptoethanol, glycerol, imidazole, fluorescein isothiocyanate (FITC), β-galactosidase (β-ga), and all solvents were purchased from Sigma-Aldrich (USA) and used as received.

**Fabrication of microwell array on silicon template**. The Si wafers were cleaned with acetone and isopropyl alcohol (IPA) and rinsed thoroughly with deionized (DI) water, followed by spin-coating with an adhesion promoter, namely HMDS, and then PR (AZ GXR-601). After soft baking at 90 °C for 90 s to remove any solvent, the wafer was exposed to ultraviolet (UV) light using a mask aligner (MA6 Mask Aligner, Suss MicroTec, Germany), followed by post-exposure baking at 110 °C for 90 s, developing in a fresh developer (AZ 300 MIF, Merck, Germany) with agitation, rinsing with DI water, and hard baking at 110 °C for 120 s. The fabrication was completed via anisotropic etching of open areas using the Bosch process (deep reactive ion etching, DRIE) and removal of the remaining PR with acetone in a bath sonicator and oxygen plasma.

**Surface modification of Si microwell array template**. The Si template (15 × 15 mm²) was cleaned with piranha solution (95% sulfuric acid: 35% hydrogen peroxide = 3:1) for 15 min and dried in an oven at 80 °C for 30 min, followed by UV/ozone treatment at a wavelength of 254 nm for 5 min for surface termination with hydroxyl groups. A PDMS stamp (20 × 20 × 10 mm³) was prepared by immersing one side in 0.1% (v/v) PFDDTS dissolved in hexane for 60 s and drying in ambient conditions for 10 min, and the PFDDTS absorbed surface was contacted with the top surface of the Si template at a pressure of 1–10 N for 1, 4, or 6 min. The Si template was rinsed with hexane and DI water and cured at 120 °C for 30 min.

**Patterning of block copolymers into microwell array**. An appropriate amount of toluene was added into a vial containing PBd-PEO with 1% (w/w) rhodamine B terminated PBd-PEO for fluorescence observation to obtain 1.5, 2.0, 2.5, and 4.0 wt% block copolymer solutions. The prepared block copolymer solution was spin-coated on the silane-treated Si templates, which were cleaned with IPA and DI water at a rate of 100 rpm for 30 s and 1000 rpm for 30 s, followed by drying in a freeze dryer for more than 6 h to remove any traces of the solvent.

**Formation of 3D block copolymer artificial cell membrane by electric field**. A micro-electro-mechanical system (MEMS) reactor to simultaneously produce, observe, and utilize the 3DBCPMs was fabricated using a PDMS soft lithography technique (detailed structure in Supplementary Fig. S8). Briefly, the bottom part of the reactor was prepared by bonding a PDMS layer (65 × 20 × 0.5 mm³) with an open area (15 × 15 mm²) for the insertion of a Si template into a glass substrate (75 × 25 × 1 mm³) by oxygen plasma treatment, and the top part of the reactor was prepared by integrating a PDMS layer (65 × 20 × 0.5 mm³) with indium tin oxide (ITO, 200 nm thick, ~50 Ω cm⁻¹) coated over glass (30 × 25 mm²) and microfluidic channels. For the experiments, the block copolymer-coated Si template was located in the open area on the bottom part, and the top and bottom parts were bonded together, where the ITO and the surface of the Si template faced each other. Upon infusion of 10 mM sucrose solution (hydration buffer) into the microfluidic channel at a rate of 20 μL/h using a syringe pump (NE-1000, New Era Pump Systems Inc., US), a sinusoidal alternating current (AC) electric field was applied between the ITO and the Si template by a function generator (33210 A, Keysight, USA). The electric field with a specific frequency of 10 Hz to 1 MHz was increased by 100 mV every 5 min until the desired voltage was reached, and it was maintained for an hour to generate all 3DBCPMs completely. To analyze the size of 3DBCPMs, we measured diameters and lengths for spherical and cilia structures, respectively, where the length for cilia structure was a distance from a microwell surface to the peak of cilia structure (Supplementary Fig. S7).

**Formation of hydrogel-supported 3DBCPMs**. A hydrogel, which is a network of hydrophilic polymer chains, can be injected into a variety of lipid and polymer structures, thereby enhancing the stability by cross-linking the polymer chains under external stimuli such as heat or light. Here, UV light was used to induce cross-linking of hydrogels that were injected immediately after the formation of cilia-like 3DBCPMs with the desired size and shape via a solution exchange. After the formation of the 3DBCPMs, the hydration buffer was exchanged with a 10 mM sucrose solution mixed with 7% PEGDMA and 5 μL of initiator (2-hydroxy-2-methyl-propiophenone) at a rate of 20 μL/h while maintaining the electric field. When the solution exchange was completed, the flow rate was set to zero, and the hydrogel solution was cured by irradiating the solution with UV light with a

wavelength of 365 nm through ITO integrated on the reactor for 10 min with a UV lamp (UVITEC, 15 W).

**Evaluation of mechanical stability and lifetime of 3DBCPMs**. Pressure gauges (EIPS345, FLUIGENT, France) were connected to the inlet and outlet of the MEMS reactor to evaluate the mechanical stability. After the formation of 3DBCPMs with different hydrogel concentrations, we changed the flow rate to regulate the pressure applied to the 3DBCPMs and detected the pressure at which the 3DBCPMs collapsed. The lifetime of the 3DBCPMs with and without 7% hydrogel was determined by counting intact structures with a confocal fluorescence microscope for 50 days.

**Expression and purification of prostaglandin E2 receptor 4**. For the preparation of the EP4 expression vector, the cDNA fragment representing amino acids 1–354 of human EP4 was amplified by polymerase chain reaction (PCR) with a forward primer (5′-TATTTT-CAGTCGACGATGGAATTCGAAACCAA CTTCTCCACTC CTCTG-3′) and reverse primer (5′-GTGATGGTGAGAAGCTTCGAATTC CATT GCCTG- TAACTCAGTCTCTGC-3′) and ligated at the EcoRI site for the P9 vector. The expression vector of P9-EP4 was transformed into E. coli strains. Freshly transformed cells were cultured in 5 mL of Luria–Bertani (LB) medium at 37 °C, supplemented with 100 μg/mL ampicillin and 30 μg/mL chloramphenicol until the optical density of the culture measured at a wavelength of 600 nm (OD₆₀₀) reached from 0.5 to 0.6. The expression of P9-EP4 was induced by adding 1 mM IPTG at 37 °C for 2 h or at 25 °C for 3 h. Cells were harvested by centrifugation at 10,000 × g for 20 min and lysed using a microfluidizer (M-110P, Microfluidics, USA). The membrane fraction was recovered from the pellet fraction after centrifugation at 100,000 × g for 1 h and resuspended in buffer (20 mM HEPES, pH 7.4) mixed with 1% sarkosyl, 10 mM β-mercaptoethanol, and 10% glycerol; the membrane proteins were solubilized by gentle agitation at 4 °C for 1 h. After washing the Ni-NTA with 5 mM imidazole in equilibration buffer, the bound proteins were eluted with 50 mM imidazole. The eluted protein fraction was incubated with amphiphilic polymer APG (amphipathic poly-γ-glutamic acid) (mass ratio of P9-EP4 and APG = 1:4) at 4 °C for 2 h and concentrated using an Amicon Ultra concentrator (Millipore, USA, 10 kDa molecular weight cutoff). The purified protein was stored at −80 °C until use.

**Reconstitution of EP4 on 3DBCPM**. Before the experiment, the MEMS reactor was treated with 3% Poloxamer 188 solution for 12 h to prevent nonspecific adsorption of proteins. A 5 mM HEPES buffer (pH 7.4) containing 14 μg/mL of the expressed P9-EP4 with APG was infused into the 3DBCPMs through a microfluidic channel at a flow rate of 20 μL/h. For the visualization of the reconstituted proteins, FITC-tagged P9-EP4 was added at a concentration of 0.1%. After protein reconstitution for 3 h, the protein solution was exchanged with 10 mM sucrose solution (or 5 mM HEPES buffer) for 30 min, and the reconstitution of membrane protein EP4 on the 3DBCPM was confirmed by measuring the fluorescence intensity of the 3DBCPM via confocal microscopy.

**Fabrication of artificial human intestinal organ**. All the reactions were performed in a MEMS reactor, and all the solutions were prepared immediately before the experiment. First, 1 mol% of DSPE-PEG2000-COOH was mixed with the block copolymer solution to incorporate β-galactosidase into the 3DBCPMs. After the formation of the 3DBCPMs, the fabrication of artificial human intestinal organs on the 3DBCPMs was completed via incubation with EDC solution (5 mM EDC, 10 mM HEPES, pH 6) for 30 min and β-ga solution (2 mg/ml β-ga, 10 mM HEPES, pH 6) for another 30 min. To confirm the digestion process of the artificial human intestinal organ, FDG solution (125 μM FDG, 10 mM HEPES, pH 6) was introduced at 5 μL/h, and the fluorescence intensity on the 3DBCPMs was recorded via confocal fluorescence microscopy.

**Imaging and characterizations**. To observe and analyze the 3DBCPMs, we generated 3DBCPMs in a MEMS reactor directly mounted on the stage of a confocal microscope (LSM 700 confocal microscope equipped with a ×40 c-apochromat (numerical aperture 1.2)) operated with ZEISS ZEN microscope software (version 2.6) or a fluorescence microscope (Zeiss LSM 5 PASCAL Zxioplan 2 microscope equipped with appropriate filter sets and ×10 Epiplan-Neofluar (numerical aperture 0.30), ×20 LD Epiplan (numerical aperture 0.40), and ×50 LD Epiplan (numerical aperture 0.50) objectives (ZEISS)). The images were captured using a Retiga6000 CCD camera (Qimaging, Surrey, BC, Canada) and Qimaging Pro software (version 7.0). Scanning electron microscopy (SEM, NOVA) measurements were carried out on the Si microwell templates to obtain the dimensions of the microwells, and atomic force microscopy (AFM, XE-100) and contact angle (Phoenix-MT(M)) measurements were performed on the surface-modified Si templates to evaluate the coverage of the silane self-assembled monolayer. Transmission electron microscopy (cryo-TEM, Tecnai F20-G2) was used to observe the morphology of the 3DBCPMs. Data analysis and plotting were made with Origin (version 2018) and Excel (version 2013) softwares.

**Statistics and reproducibility**. Figures 1c, 2b, d, and 4b and supplementary Figs. 9 and 11a–c are representative of more than 10 images obtained from individual experiments. All results and experiments were consistent and reproducible.

**Reporting summary**. Further information on research design is available in the Nature Research Reporting Summary linked to this article.

## Data availability

All data supporting the findings and conclusions of this study are available within the paper and its Supplementary Information files. All other relevant data are available from the corresponding author upon reasonable request. Source data are provided with this paper.

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

## Acknowledgements

This work was supported by the Korean Medical Device Development Fund grant funded by the Korean government (the Ministry of Science and ICT, the Ministry of Trade, Industry and Energy, the Ministry of Health & Welfare, the Ministry of Food and Drug Safety) (9991006807, KMDF_PR_20200901_0134_2021_01), supported by the National Research Foundation of Korea (NRF) grant funded by the Korea government (MIST) (NRF-2020R1A2C2100363), and also supported by KIST Institutional Program (2E31502).

## Author contributions

D.-H.K., W.B.H., and T.S.K. conceived and designed the research. D.-H.K., W.B.H., H.I.R., and T.Y.K. performed the experiments. N.H.K. and Y.G.Y. contributed to the protein preparation. N.C. and J.Y.K. supported the experiments and commented on the paper. D.-H.K., W.B.H., and T.S.K. co-wrote the manuscript.

## Competing interests

The authors declare no competing interests.
