## [Peer Review File · Nature Communications]

REVIEWER COMMENTS

Reviewer #1 (Remarks to the Author):

Kim and co-worker report here on a tunable/scalable fabrication method for block copolymer-based 3D artificial cell membrane array.

The results here are indeed impressive. The robustness and homogeneity of the structures formed is remarkable. Given the importance of having topological control over the self-assembly of membrane-like structures from block-polymers, this work is likely to have a significant impact.

I will though flag that this paper relies very heavily on using electric fields and various microfluidic methods to create and study these structures. This is too removed from own specialty, which is in self-assembled structures in solution, for me to be able to give a comprehensive opinion on this work; both from the point of view how novel it is (it seems to me) or if some the methodology with regards to microfluidics/fabrication and use of electric fields is wanting. I hope therefore that other reviewers are better place to make an authoritative judgement on those matters.

I do nevertheless have a couple of minor points / suggestions/ questions:

1. The authors should consider reducing their reliance on abbreviation and perhaps replace some of them with the full name. It won't make the manuscript much longer but it will make it a lot more readable (abbreviations are truly the enemy of clarity in much of the nanoscience and material chemistry literature).

2. Line 240 "amphipathic poly r-glutamic acid" What is the "r" here? Should this be a small caps L? Or a gamma?

3. Lines 270-275 – about Michael-Menten's mechanism – the argument used / in the SI appears circular: As explained in the SI, if one assumes [S] is much larger than K(M) (the Michael-Menten's constant) then yes, the Michael-Menten's equation will simply down to $v_0 = V_{max} = k_{act}[E_0]$ – which is functionally just like any other first order rate equation. But to then use the argument that since the data (see lines 267-269) fits a first order equation, there a Michael-Menten mechanism is valid seems like a circular reasoning. Please consider changing this.

Reviewer #2 (Remarks to the Author):

This manuscript describes electric field-assisted 3-dimensional BCP artificial cell membranes (3DBCPMs) with different morphologies controlled by the amplitude and frequency of the electric field. Meanwhile, the two electric parameters (i.e., amplitude and frequency) predetermine the general 3D shape of BCP membranes growing from predefined silicon microwells such as sphere and cilia structures, changing the concentration of the BCP controls the aspect ratio (between length and diameter) of the 3DBCPMs. Finally, the authors applied this scalable, controllable formation of 3DBCPMs for biological applications using a protein-membrane interaction assay by showing the most rapid increase in fluorescence intensity with time on cilia-shaped BCP membranes, compared with sphere and planar membranes. Overall, I found some interest in this work and the manuscript is

written very nicely. However, the same group already demonstrated the electric-field-assisted 3D BCP membrane formation in *Advanced Materials Interfaces* in 2019 (*Adv. Mater. Interfaces* 2019, 6, 1801554), which the authors indicated in the main text. Compared to the previous one, the major advance in this manuscript is the better controllable and scalable fabrication of 3DBCPMs in a more precise manner in terms of amplitude and electric field and employed a different biological assay as a proof-of-concept for biological applications. Based on my assessment, this manuscript does not seem to be very new, like the first demonstration of electric field-assisted self-assembly of 3DBCPMs as the authors claim. There are some points needed to be reconsidered by the authors.

1. In line 152, the Maxwell-Wagner time scale (t_{MW}) is shown. Why is the different equation employed compared to the general form of MW? Specifically, ϵ_{ex} in this manuscript, $2\epsilon_{ex}$ in the original equation and other literature. Please explain.

2. In lines 166-167, the authors claim that the 3DBCPM should have a spherical shape at a frequency < 1 kHz, and spherical shapes were observed in Figure 3b independent of BCP concentrations at low frequencies. However, in general, at low frequencies ($f < 10^3$ - 10^4 Hz), vesicles experience morphological transitions in response to an external electric field due to the charges developed on the membrane boundary, forming prolate shapes aligned with electric field direction (*Soft Matter*, 2009, 5, 3201–3212; *Soft Matter*, 2007, 3, 817–827; *Biophysical Journal: Biophysical Letters*, 2008, 95, PL19-21, etc.). The authors should comment on this different observation.

3. Please denote the time scale for those experiments shown in Figure 3. To produce/observe different aspect ratios, the time scale (duration) for all the experiments should be the same for fair comparison and might be a critical factor.

4. In addition, membrane deformation or disruption would be caused by different time scales, amplitude, and frequency. Please mention about this disruption possibility or add experimental observations.

Tunable and scalable fabrication of block copolymer-based 3D polymorphic artificial cell membrane array

D.-H. Kang et al.

This manuscript describes electric field-assisted 3-dimensional BCP artificial cell membranes (3DBCPMs) with different morphologies controlled by the amplitude and frequency of the electric field. Meanwhile, the two electric parameters (i.e., amplitude and frequency) predetermine the general 3D shape of BCP membranes growing from predefined silicon microwells such as sphere and cilia structures, changing the concentration of the BCP controls the aspect ratio (between length and diameter) of the 3DBCPMs. Finally, the authors applied this scalable, controllable formation of 3DBCPMs for biological applications using a protein-membrane interaction assay by showing the most rapid increase in fluorescence intensity with time on cilia-shaped BCP membranes, compared with sphere and planar membranes. Overall, I found some interest in this work and the manuscript is written very nicely. However, the same group already demonstrated the electric-field-assisted 3D BCP membrane formation in *Advanced Materials Interfaces* in 2019 (*Adv. Mater. Interfaces* 2019, 6, 1801554), which the authors indicated in the main text. Compared to the previous one, the major advance in this manuscript is the better controllable and scalable fabrication of 3DBCPMs in a more precise manner in terms of amplitude and electric field and employed a different biological assay as a proof-of-concept for biological applications. Based on my assessment, this manuscript does not seem to be very new, like the first demonstration of electric field-assisted self-assembly of 3DBCPMs as the authors claim. There are some points needed to be reconsidered by the authors.

1. In line 152, the Maxwell-Wagner time scale (t_{MW}) is shown. Why is the different equation employed compared to the general form of MW? Specifically, ϵ_{ex} in this manuscript, $2\epsilon_{ex}$ in the original equation and other literature. Please explain.
2. In lines 166-167, the authors claim that the 3DBCPM should have a spherical shape at a frequency < 1 kHz, and spherical shapes were observed in Figure 3b independent of BCP concentrations at low frequencies. However, in general, at low frequencies ($f \leq 10^3$ - 10^4 Hz), vesicles experience morphological transitions in response to an external electric field due to the charges developed on the membrane boundary, forming prolate shapes aligned with electric field direction (Soft Matter, 2009, 5, 3201–3212; Soft Matter, 2007, 3, 817–827; Biophysical Journal: Biophysical Letters, 2008, 95, PL19-21, etc.). The authors should comment on this different observation.
3. Please denote the time scale for those experiments shown in Figure 3. To produce/observe different aspect ratios, the time scale (duration) for all the experiments should be the same for fair comparison and might be a critical factor.
4. In addition, membrane deformation or disruption would be caused by different time scales, amplitude, and frequency. Please mention about this disruption possibility or add experimental observations.

Response to the reviewer 1's comments

Kim and co-worker report here on a tunable/scalable fabrication method for block copolymer-based 3D artificial cell membrane array.

The results here are indeed impressive. The robustness and homogeneity of the structures formed is remarkable. Given the importance of having topological control over the self-assembly of membrane-like structures from block-polymers, this work is likely to have a significant impact.

I will though flag that this paper relies very heavily on using electric fields and various microfluidic methods to create and study these structures. This is too removed from own specialty, which is in self-assembled structures in solution, for me to be able to give a comprehensive opinion on this work; both from the point of view how novel it is (it seems to me) or if some the methodology with regards to microfluidics/fabrication and use of electric fields is wanting. I hope therefore that other reviewers are better placed to make an authoritative judgement on those matters.

[RESPONSE] We appreciate the reviewer for this positive evaluation.

I do nevertheless have a couple of minor points / suggestions/ questions:

1. The authors should consider reducing their reliance on abbreviation and perhaps replace some of them with the full name. It won't make the manuscript much longer but it will make it a lot more readable (abbreviations are truly the enemy of clarity in much of the nanoscience and material chemistry literature).

[RESPONSE] We agree with the reviewer's comment. In order to improve the clarity, we removed the abbreviations, such as BCP for block copolymer, CA for contact angle, GPCR for G-protein coupled receptor, and PEB for post-exposure baking, in the manuscript and figures.

[MODIFICATION] We added the full name of some abbreviations in the manuscript as below.

"This behavior has been widely exploited in 2D/3D nanoscale patterning"

To

"This behavior has been widely exploited in 2-dimensional (2D)/3-dimensional (3D) nanoscale patterning"

"..... DNA-directed self-assembly have been reported"

To

"..... deoxyribonucleic acid (DNA)-directed self-assembly have been reported"

2. Line 240 "amphipathic poly r-glutamic acid" What is the "r" here? Should this be a small caps L? Or a gamma?

[RESPONSE] It was a typo. We corrected "r" to "-γ" in the manuscript and figure caption.

[MODIFICATION]

"Here, amphipathic poly r-glutamic acid (APG) and green fluorescent protein (GFP) were used"

To

“Here, amphipathic poly- γ -glutamic acid (APG) and green fluorescent protein (GFP) were used

“The eluted protein fraction was incubated with amphiphilic polymer APG (amphipathic poly γ -glutamic acid)

To

“The eluted protein fraction was incubated with amphiphilic polymer APG (amphipathic poly- γ -glutamic acid)

“..... stabilized with amphipathic poly- γ -glutamic acid (APG):

To

“..... stabilized with amphipathic poly- γ -glutamic acid (APG):

3. Lines 270-275 – about Michael-Menten’s mechanism – the argument used / in the SI appears circular: As explained in the SI, if one assumes [S] is much larger than $K(M)$ (the Michael-Menten’s constant) then yes, the Michael-Menten’s equation will simply down to $v_0 = V_{max} = k_{cat}[E]_0$ – which is functionally just like any other first order rate equation. But to then use the argument that since the data (see lines 267-269) fits a first order equation, there a Michael-Menten mechanism is valid seems like a circular reasoning. Please consider changing this.

[RESPONSE] Thank you for your comment. We use the Michaelis-Menten equation to double-check the reaction rate of enzyme. Since the [S] was very high in our experiments, we simplified the Michaelis-Menten equation as $v_{initial} \approx V_{max} = k_{cat}[E]_0$. The enzyme (β -Ga) bound to EDC, which was conjugated to 1% DSPE-PEG-NHS in 3DBCPMs; thus, the [E] was proportional to the surface area of 3DBCPMs. Due to the identical k_{cat} in the same kind of enzymatic reaction, we can calculate the relative V_{max} values for different shapes of 3DBCPMs. On the other hand, we derived $v_{initial}$ from the slope of the initial rate period of experimentally obtained Figure 5e. As shown in Fig. S12, the reaction rate from the experimental result ($v_{initial}$, black squares) higher than the calculated one (V_{max} , red hollow circles), which means that this behavior was originated from the 3D architecture and large surface area of the spherical and cilia structures proposed in this work. Such behaviors were favorable for active and effective reactions. This whole process was independent of the first-order reaction, and the Michaelis-Menten equation was just used for the view of enzymatic reaction. We reconfirmed intestinal organ mimicking as different of view using the Michaelis-Menten equation. Thus, it is not circular reasoning.

[MODIFICATION] To enhance the understanding of the reader, we added and revised some sentences in the SI as below.

“..... in the same kind of enzymatic reaction, V_{max} depended on $[E]_0$.”

To

“..... in the same kind of enzymatic reaction, V_{max} depended on $[E]_0$. Thus, we can calculate proportional V_{max} followed different types of 3DBCPMs using surface area of them as shown in Fig. S12 (red hollow circle).”

“And, the [P] was associated with the fluorescence intensity as shown in Fig. 5e, so the slope of the initial rate period could be expressed as $v_{initial}$ (Fig. S12).”

To

“And, the [P] was associated with the fluorescence intensity as shown in Fig. 5e, so the slope of the initial rate period in Fig. 5e could be expressed as v_{initial} , which is derived by experimental results (Fig. S12).”

Response to the reviewer 2's comments

This manuscript describes electric field-assisted 3-dimensional BCP artificial cell membranes (3DBCPMs) with different morphologies controlled by the amplitude and frequency of the electric field. Meanwhile, the two electric parameters (i.e., amplitude and frequency) predetermine the general 3D shape of BCP membranes growing from predefined silicon microwells such as sphere and cilia structures, changing the concentration of the BCP controls the aspect ratio (between length and diameter) of the 3DBCPMs. Finally, the authors applied this scalable, controllable formation of 3DBCPMs for biological applications using a protein-membrane interaction assay by showing the most rapid increase in fluorescence intensity with time on cilia-shaped BCP membranes, compared with sphere and planar membranes. Overall, I found some interest in this work and the manuscript is written very nicely. However, the same group already demonstrated the electric-field-assisted 3D BCP membrane formation in *Advanced Materials Interfaces* in 2019 (*Adv. Mater. Interfaces* 2019, 6, 1801554), which the authors indicated in the main text. Compared to the previous one, the major advance in this manuscript is the better controllable and scalable fabrication of 3DBCPMs in a more precise manner in terms of amplitude and electric field and employed a different biological assay as a proof-of-concept for biological applications. Based on my assessment, this manuscript does not seem to be very new, like the first demonstration of electric field-assisted self-assembly of 3DBCPMs as the authors claim.

[RESPONSE] We appreciate the reviewer's evaluation on our work. However, we think there is a misunderstanding about our previous paper. Our previous paper (*Adv. Mater. Interfaces* 2019, 6, 1801554) focused on the generation of 3D structures based on 'phospholipid' not 'block copolymer' by using an electric field. Physical/biological properties are quite different between phospholipids and block copolymers, and block copolymer-based artificial cell membranes have many advantages compared to phospholipid-based counterparts, as mentioned in the manuscript. Therefore, our work on the controllable and scalable fabrication of 3D block copolymer structures and in vitro biological assays can't only be considered as new findings but also have significance in various biotechnological applications.

There are some points needed to be reconsidered by the authors.

1. In line 152, the Maxwell-Wagner time scale (t_{MW}) is shown. Why is the different equation employed compared to the general form of MW? Specifically, ϵ_{ex} in this manuscript, $2\epsilon_{ex}$ in the original equation and other literature. Please explain.

[RESPONSE] It was a typo. We corrected " ϵ_{ex} " to " $2\epsilon_{ex}$ " in the manuscript.

[MODIFICATION]

$$t_{MW} = \frac{\epsilon_{in} + \epsilon_{ex}}{\lambda_{in} + 2\lambda_{ex}}$$

To

$$t_{MW} = \frac{\epsilon_{in} + 2\epsilon_{ex}}{\lambda_{in} + 2\lambda_{ex}}$$

2. In lines 166-167, the authors claim that the 3DBCPM should have a spherical shape at a frequency < 1 kHz, and spherical shapes were observed in Figure 3b independent of BCP concentrations at low frequencies. However, in general, at low frequencies ($f < 10^3$ - 10^4 Hz), vesicles experience morphological transitions in response to an external electric field due to the charges developed on the membrane boundary, forming prolate shapes aligned with electric field direction (*Soft Matter*, 2009, 5, 3201-3212; *Soft Matter*, 2007, 3, 817-827; *Biophysical Journal: Biophysical Letters*, 2008, 95, PL19-21, etc.). The authors should comment on this different observation.

[RESPONSE] As phospholipid-based vesicles in the papers you mentioned, our 3DBCPMs experience morphological transitions at a frequency of ~ 1 kHz, forming prolate shapes. However, the aspect ratio of the 3DBCPMs was experimentally and theoretically observed as < 1.1 and < 1.3 , respectively. These values are far

less than 1.5, the boundary value between 'spherical shape' and 'cilia shape' defined for structural comparison in this study, so we claimed that 3DBCPMs generated at a frequency $< 1\text{kHz}$ are spherical shapes. To avoid confusion, we defined 'spherical shape' and 'cilia shape' in the manuscript.

[MODIFICATION] We added/modified the sentence to the manuscript as below

"..... between 1 kHz and 1 MHz (Fig. 3b). The experimental results are consistent with such a theoretical estimation."

To

"..... between 1 kHz and 1 MHz (Fig. 3b). Here, we defined 3DBCPMs with aspect ratios below and above 1.5 as 'spherical shape' and 'cilia', respectively, for structural distinction. The experimental results (aspect ratio, < 1.1) are nearly consistent with such a theoretical estimation (aspect ratio, < 1.3)."

We removed the following sentence from the manuscript.

"Here, cilia refer to 3DBCPMs with an aspect ratio of > 1.5 ."

3. Please denote the time scale for those experiments shown in Figure 3. To produce/observe different aspect ratios, the time scale (duration) for all the experiments should be the same for fair comparison and might be a critical factor.

[RESPONSE] We agree with the reviewer's comment that the time scale is an important factor for the fabrication of 3DBCPMs. In this work, we applied an electric field by 100 mV every 5 min until the desired voltage was reached (to avoid any deformation caused by a sharp increase in the electric field) and maintained the voltage for an hour to complete the generation of 3DBCPMs. This time was enough to produce all 3DBCPMs.

[MODIFICATION] To avoid confusion and aid understanding, we modified the following sentences in the manuscript.

"..... it was maintained to generate 3DBCPMs. The whole process for the generation of 3DBCPMs took about an hour on average."

To

"..... it was maintained for an hour to generate all 3DBCPMs completely."

We added the following sentence in the figure caption for Figure 3.

"..... dynamics of 3DBCPM."

To

"..... dynamics of 3DBCPM. All 3DBCPMs were produced for an identical time of an hour (except for the time to increase an amplitude by 100 mV every 5 min until the amplitude reached the desired value)."

4. In addition, membrane deformation or disruption would be caused by different time scales, amplitude, and frequency. Please mention about this disruption possibility or add experimental observations.

[RESPONSE] We thank the reviewer for the detailed comment. The generated 3DBCPMs could transform slowly in response to different amplitudes and frequencies and lost their structure aligned along the direction of the electric field when the electric field was removed, as shown in Fig 4c. Nevertheless, the 3DBCPMs could maintain very stably for several days due to the robustness of block copolymers. Although membrane deformation or disruption gradually occurred over time, more than 80% of 3DBCPMs still survived after 50 days. Such outstanding stability is a huge advantage of 3DBCPMs proposed in this study. Furthermore, the introduction of hydrogel to support 3DBCPMs significantly enhances the stability (~ 100% survival) over 50 days regardless of the electric field conditions, as shown in Fig. 4e.

[MODIFICATION] We added the following sentences to the manuscript.

“..... practical use as biological platforms. However, unlike spherical structures,”

To

“..... practical use as biological platforms. However, the generated 3DBCPMs slowly transform their original morphology into a different morphology in response to different amplitudes and frequencies. Furthermore, unlike spherical structures,”

“To circumvent this issue and ensure not only”

To

“To circumvent these issues and ensure not only”

“Despite the slight changes in morphology, the final structure still had a high aspect ratio and excellent mechanical stability.”

To

“Despite the slight changes in morphology, the final structure still had a high aspect ratio and excellent mechanical stability regardless of the electric field conditions.”

REVIEWERS' COMMENTS

Reviewer #1 (Remarks to the Author):

The authors have addressed all my comments and suggestions. I am happy to support the publication of this manuscript.

Reviewer #2 (Remarks to the Author):

The authors addressed all my comments. Thus, I would like to recommend this manuscript for publication.